# COVID-19 isolation and quarantine orders in Berlin-Reinickendorf (Germany): How many, how long and to whom?

**Jakob Schumacher**[1,2]*, **Lisa Kühne**[3], **Sophia Brüssermann**[3], **Benjamin Geisler**[4], **Sonja Jäckle**[4]

1 Local Public Health Agency, Berlin, Germany, 2 Robert Koch Institute, Berlin, Germany, 3 Leibniz Institute for Prevention Research and Epidemiology - BIPS, Bremen, Germany, 4 Fraunhofer Institute for Digital Medicine MEVIS, Lübeck, Germany

* schumacherj@rki.de

**Data Availability Statement:** All code and datafiles are available from the github repository https://github.com/jakobschumacher/quarantine-isolation-analysis.

## Abstract

Isolating COVID-19 cases and quarantining their close contacts can prevent COVID-19 transmissions but also inflict harm. We analysed isolation and quarantine orders by the local public health agency in Berlin-Reinickendorf (Germany) and their dependence on the recommendations by the Robert Koch Institute, the national public health institute. Between 3 March 2020 and 18 December 2021 the local public health agency ordered 24 603 isolations (9.2 per 100 inhabitants) and 45 014 quarantines (17 per 100 inhabitants) in a population of 266 123. The mean contacts per case was 1.9. More days of quarantine per 100 inhabitants were ordered for children than for adults: 4.1 for children aged 0-6, 5.2 for children aged 7-17, 0.9 for adults aged 18-64 and 0.3 for senior citizens aged 65-110. The mean duration for isolation orders was 10.2 and for quarantine orders 8.2 days. We calculated a delay of 4 days between contact and quarantine order. 3484 contact persons were in quarantine when they developed an infection. This represents 8% of all individuals in quarantine and 14% of those in isolation. Our study quantifies isolation and quarantine orders, shows that children had been ordered to quarantine more than adults and that there were fewer school days lost to isolation or quarantine as compared to school closures. Our results indicate that the recommendations of the Robert Koch Institute had an influence on isolation and quarantine duration as well as contact identification and that the local public health agency was not able to provide rigorous contact tracing, as the mean number of contacts was lower than the mean number of contacts per person known from literature. Additionally, a considerable portion of the population underwent isolation or quarantine, with a notable number of cases emerging during the quarantine period.

## Introduction

Separation orders for cases (isolation) and for contacts (quarantine) are public health interventions to slow down the spread of COVID-19. Both are recommended by the WHO [1] and

**Funding:** Jakob Schumacher received no specific funding for this work. Lisa Kühne, Sophia Brüssermann, Benjamin Geisler and Sonja Jäckle were funded by the German Federal Ministry of Education and Research (BMBF, Project EsteR, Funding Code: 13GW0542).

**Competing interests:** The authors have declared that no competing interests exist.

both are serving as an integral part of the COVID-19 response [2]. While isolation and quarantine orders are critical in controlling the spread of COVID-19, they are not without adverse effects. These measures involve restricting individuals' freedom, often leading to psychological distress, social isolation, and broader socio-economic challenges [3–5]. It is essential to weigh these drawbacks against the benefits in the overall strategy against COVID-19. Other important measures where school closures, that were mandated by the federal states. In Berlin, schools were closed for 124 days, spanning from March 13 to May 29, 2020, and January 6 to February 22, 2021.

COVID-19 is caused by SARS-CoV-2, a virus that emerged in 2019 in China, likely by a zoonotic spillover [6]. Since then it has spread all over the world and caused a pandemic. The disease is spread from human to human. It is a respiratory disease with a pathway via the ACE-2-receptor, that can range from asymptomatic cases to severe respiratory infections and death [7]. Vaccines have been developed and proven successful in inducing seroconversion [8], preventing severe disease and death [9], but vaccine hesitancy hinders the effectiveness [10]. Novel drugs have been developed, some of which are useful, especially for individuals with a high risk [11].

The legal framework in Germany for isolation and quarantine orders is outlined in § 28–30 of the infection protection act. The national institute for public health, the Robert Koch Institute, recommends a duration for the isolation and quarantine period and provides a definition for contact persons [12, 13]. The 16 federal states of Germany have legal frameworks for contact tracing that usually follow the recommendations of the Robert Koch Institute. There are 378 local public health agencies that execute the isolation and quarantine orders for cases and contacts, respectively. Some local public health agencies, including Berlin-Reinickendorf, issued a general decree for isolation and quarantine, following the national and federal state laws. The local public health agency of Berlin-Reinickendorf established an internal guideline document ("Reinickendorfer Coronavirus Update") that harmonised the work of the staff and specified the aforementioned law documents.

The local public health agency Berlin-Reinickendorf is responsible for a district of Berlin with about 1/4 million inhabitants. Before the COVID-19 pandemic, the infectious disease unit of the agency consisted of 9 persons. During the pandemic, up to 140 persons were working on contact tracing and case isolation at a time. The additional staff consisted mainly of staff from district and state agencies, containment scouts from an initiative by the Robert Koch Institute and soldiers from the German military force. The incidence of COVID-19 in Berlin-Reinickendorf followed the wave pattern for Germany as a whole. Due to the high work load the agency was not able to uphold contact tracing starting from December 2021. The local public health agencies of Berlin were working in close collaboration: throughout most of the time, the local public health agency of Berlin-Reinickendorf managed all contacts from Berlin, that resulted from cases in Berlin-Reinickendorf, but when a contact turned positive, the case was managed in the district where the person was living.

Available literature that directly analyse field work on isolation and quarantine orders are scarce. In the United States of America, two studies analysed a student initiative that traced 953 contacts of 536 COVID-19 cases around Pennsylvania [14, 15] and Sachdev et al. analysed the tracing of 1214 contacts of 1633 cases in San Francisco [16]. Shi et al. analysed 183 cases and their 1983 close contacts in Whanzou, China in much detail [17] and Jian et al. reported on the Taiwanese digital system TRACE analysing 487 cases and 8051 contact persons [18]. Public health implications of these systems included surveillance of travel history [19] and ensuring an adequate quantity of personal protective equipment [20]. Mossong et al. traced 2721 contacts of 424 cases in schools in Luxemburg, resulting in an average quarantine duration of 4.3 days for pupils. The proportion of quarantined persons that turned out to be

secondary cases depended on the setting where the transmission took place. Within schools, 2.2% of pupils and 1.1% of teachers got infected after contact to a case, while transmission within families occurred for 14% of contacts [21]. According to existing literature, the average person in Germany typically has about 7.95 contacts per day [22].

The goal of this work is to quantify and analyse isolation and quarantine orders from a local public health agency in Germany to assess differences among age groups, mean number of contacts, timeliness of quarantine orders, the number of contained or non-contained cases and to estimate the impact of the contact person tracing. In addition, we evaluate the influence of the recommendations of the Robert Koch Institute on the number and duration of isolation and quarantine orders. The results are intended to support decisions by health authorities regarding the revisions of recommendations for current and future outbreaks. To the best of our knowledge, this is the first study in Germany that analyses field work on isolation and quarantine orders and the largest analysis worldwide that covers nearly two years of the pandemic.

## Methods

### Contact tracing and quarantine strategy

After the notification of a case the staff of the local public health agency contacted the case and identified individuals who had been in close contact with the case, where the definition of "close contact" was taken from the recommendations of the Robert-Koch-Institute. Once identified, these contacts were placed under quarantine. During the quarantine period the individuals were instructed to regularly check and report symptoms indicative of COVID-19. To optimize the use of testing resources for conducting PCR tests, the staff employed a scoring system, termed 'Abstrichscore.' This system prioritized individuals based on several criteria, with contact persons typically receiving the highest scores, especially when exhibiting symptoms. Consequently, during the study persiod nearly all contact persons were selected for testing.

### Data retrieval and cleaning, inclusion and exclusion criteria and handling of missing data

The population under study consisted of all people residing in Berlin Reinickendorf and had a separation order in the study period. The study period ranged from the time of notification of the first case, the 3 March 2020, until the 18 December 2021, when the local public health agency was unable to continue recording contacts due to the high number of cases. Our data on isolation and quarantine orders were retrieved on 10 February 2022 from the database of separation orders of the reporting software of the agency. (SurvN et@RKI [23]). The following variables were extracted from the list of isolation orders as well as the quarantine orders: beginning of separation order, end of separation order, reporting date, age group (0–6, 7–17, 18–59, 60–110 years). We generated a person ID in Microsoft Excel consisting of name, surname, date of birth and the address directly after the export. Personal data and the links to the person ID were subsequently deleted and the—now anonymous—person ID and the remaining data was exported to be used in the programming language R [24]. Entries with missing values in one of the date variables for separation orders were not included. We defined a separation order as a record in the study period with an anonymous person ID that indicate an existing person. Entries with presumed typing errors in one of the date variables were removed. A typing error was assumed in case 1) any date was not between 3 March 2020 and 1 January 2022 (the end of the study period plus 14 days), 2) the separation order was longer than 30 days or

less than one day or, 3) the beginning of the separation order was more than 182 days (half a year) before the reporting date or more than 182 days after the reporting date. We merged double entries when a person was assumed to have two overlapping isolation periods or two overlapping quarantine periods. In this case, the beginning of the combined period was set to the earlier date of the two start dates and the end of the combined period was set to the later of the two end dates. The non-merged entry was removed as a duplicate. If a quarantine period had an overlap with an isolation period, we set the end of a quarantine period to the date of the beginning of the isolation period.

We retrieved demographic data from the Open Data Portal Berlin. We used the latest available data from the year 2020 [25]. We retrieved the total number ($N_{all}$), as well as the number of kindergarten children aged 0 to 6 ($N_{0-6}$), school children aged 7 to 17 ($N_{7-17}$), adults aged 18 to 64 ($N_{18-64}$) and senior citizens aged 65 to 110 ($N_{64-110}$).

## Analysis of recommendations

We analysed relevant changes in the definition of contact persons [13] (contact person definition period or C), changes in the recommended duration of isolation (isolation duration period or I) and changes in the recommended duration of quarantine (quarantine duration period or Q) [12] during the study period. The data were retrieved from the website of the Robert Koch Institute, the archives of the website of the Robert Koch Institute on the Waybackmachine and the internal guideline document ("Reinickendorfer Coronavirus Update"). We considered changes as relevant when they were mentioned in the guidelines of the local public health agency or were deemed as important by the staff thereof.

## Statistical measures

We calculated the following measurements

- *Quantity of isolation and quarantine orders:* The total number of isolation $n_i$ and quarantine $n_q$ orders (in total, by age group and by contact person definition period). The number of isolation orders per 100 inhabitants $n_{i-p100}$ and quarantine orders per 100 inhabitants $n_{q-p100}$ (in total and by age group). We counted how many separation orders were given for an individual $n_{per-individual}$.

- *Duration of isolation and quarantine orders:* The mean ($\tilde{d}_i$) and median ($d_i$) duration and the interquartile range of isolation and quarantine orders ($\tilde{d}_q$, $d_q$, respectively) in total, by age group and by isolation duration or quarantine duration. We also calculated the number of days spent in isolation or quarantine order per inhabitant ($nd_{i-pi}$ or $nd_{q-pi}$) in total and by age group. We calculated the duration as the difference between the end of the separation order and the beginning of the separation order plus one day, because the separation order included the last day.

- *Ratio of quarantines to isolation orders:* We divided the number of quarantine orders by the number of isolation orders $r_{qi}$ in total or by period of recommendation for the definition of contact persons. For this calculation, we used only entries where the beginning of the separation order was after the 24 May 2020, because contact persons were not recorded in Surv*Net* prior to that date. Note that we were not able to link a contact person directly to the case with whom she or he was in contact, as the agency did not record these links in the software.

- *Number of quarantine orders that had a following isolation order:* For separation orders issued after May 24, 2020, we categorized cases into three scenarios: 1) Isolation orders immediately following quarantine, termed 'contained cases' $n_{cc}$ e.g. a person that developed

COVID-19 while being in quarantine—2) Isolation orders within 7 days post-quarantine, labeled 'non-contained cases'—we refer to this as a non-contained case $n_{ncc}$, e.g. a person that was in quarantine, got out of quarantine and then developed COVID-19 within 7 days. We chose the cut-off value of 7 because it marks half of the incubation period or 3) persons with no isolation order in the seven days following a quarantine order, retrospectively unnecessary quarantines. We calculated these numbers overall and stratified by age group, contact person definition period and quarantine duration periods.

- *Timeliness of isolation and quarantine orders:* For periods with a recommended quarantine duration of 14 days we approximated the time delay between the last day of contact and the beginning of the quarantine order $\tilde{d}_{delay}$. We subtracted the length of the quarantine order from 14 days.

$$\tilde{d}_{delay} = 14 - \tilde{d}_q \tag{1}$$

The 'last contact' refers to the date when an individual was last at risk of infection, as determined by the local public health staff. In houshold situations, this was marked by the date when the health agency staff advised the individual to start separating from the infected person.

## Reproducibility, ethics statement and data protection

The corresponding script as well as the anonymous data set are available on Github [26]. This work was conducted as part of the surveillance work of the local public health agency. Institutional review board approval and informed consent were not required. Data protection approval was given by the local agency data protection unit.

## Results

### Results of data retrieval and cleaning

We extracted 109 087 database entries from Surv*Net*. We removed several entries that did not fulfil the case definition. These included 11 215 entries with missing dates, 108 entries with an invalid ID and 24 030 entries where the separation order did not begin within the study period. We also removed 377 entries because they had a presumed typing error in one of the dates as well as 30 duplicated isolation orders and 2498 duplicated quarantine orders. We analysed 70 829 entries. For a graphical overview see S1 Fig. For 3484 quarantines we reduced the length by the overlap with a following isolation period. In the demographic data we found 266 123 inhabitants registered in Berlin-Reinickendorf.

### Results of the analysis of recommendations

We analysed changes in case definitions, isolation and quarantine duration both in the recommendations given by the Robert Koch Institute and the internal guidelines of the local public health agency Berlin-Reinickendorf during the study period. We identified three periods with relevant differences in isolation duration (denoted as $I_{1-3}$), four periods with relevant differences in quarantine duration (denoted as $Q_{1-4}$) and three periods with relevant differences in contact person definition (denoted as $C_{1-3}$). The internal guideline document closely followed the recommendations of the Robert Koch Institute, except for one change in contact person definition regarding schools and kindergartens. Main differences for isolation and quarantine duration included the length of the recommended time period and whether a negative test

**Table 1. Time periods of relevant recommendations of the Robert Koch Institute for isolation duration, quarantine duration and contact person definition.**

| Isolation duration period | | | |
|---|---|---|---|
| | from | until | recommended duration |
| $I_1$ | 03.03.2020 | 01.07.2020 | 14 days |
| $I_2$ | 02.07.2020 | 30.03.2021 | 10 days (with PoC) |
| $I_3$ | 31.03.2021 | 18.12.2021 | 14 days |
| **Quarantine duration period** | | | |
| | from | until | recommended duration |
| $Q_1$ | 03.03.2020 | 30.11.2020 | 14 days |
| $Q_2$ | 01.12.2022 | 15.02.2021 | 10 days (with PoC) |
| $Q_3$ | 16.02.2021 | 08.09.2021 | 14 days |
| $Q_4$ | 09.09.2021 | 18.12.2021 | 7 days (with PoC); 5 days (with PCR/children with PoC) |
| **Contact person definition period** | | | |
| | from | until | time definition + Group/individual + Vacc. status |
| $C_1$ | 03.03.2020 | 30.03.2021 | 15 min. + children with GQ |
| $C_2$ | 31.03.2021 | 19.05.2021 | 10 min. + children with GQ + not vacc. |
| $C_3$ | 20.05.2021 | 18.12.2021 | 10 min. + children with IQ + not vacc. |

PoC = Point of care or antigen test, PCR = Polymerase-chain-reaction test, GQ = group quarantine, where usually a complete group of children was ordered to quarantine, IQ = individual quarantine where usually individual children with direct contact were ordered to quarantine, not vacc. = vaccinated or recovered persons were not ordered to quarantine.

could be used to end the isolation or quarantine early, either with an antigen or a PCR test. The main differences in the contact person definition were the length of the face-to-face contact that led to a separation order, the way of regarding children in large groups as contact persons (group quarantine, e.g., the whole class was ordered to quarantine or individual quarantine, e.g., only the persons with a direct contact were ordered to quarantine) and if vaccinated or recovered persons were given a separation order. An overview of the time periods is shown in Table 1 and with more detail and with an excerpt from the original source and corresponding links in the S1 Table.

## Results of statistical measures

**Analysis of quantity of isolation and quarantine orders.** The local public health agency ordered $n_i$ = 24 433 isolations and $n_q$ = 45 335 quarantines ($n_{i\text{-}p100}$ = 9.2 isolation orders and $n_{q\text{-}p100}$ = 17 quarantine orders per 100 inhabitants). The distribution over time is shown in Fig 1. The number of isolation and quarantine orders by age group and contact person definition period can bee seen in Table 2. The number of quarantines per 100 inhabitants $n_{q\text{-}p100}$ was 50.6 for kindergarten children and 64.9 for school children as compared to 10.5 in adults and 3.2 in the senior citizens group. Thus school children were 20.3 times more likely to have been in quarantine than senior citizens. 46 817 (81.5%) persons had one separation order (either isolation or quarantine), 9 061 (15.8%) had two separation orders, 1 359 (2.4%) had three separation orders, 163 (0.3%) had four separation orders and 20 had five separation orders—which was the maximum number of separation orders per person.

**Analysis of the duration of isolation and quarantines.** Overall, the public health agency ordered 684 years of isolation and 1 031 years of quarantine or 1 714 years separation in total.

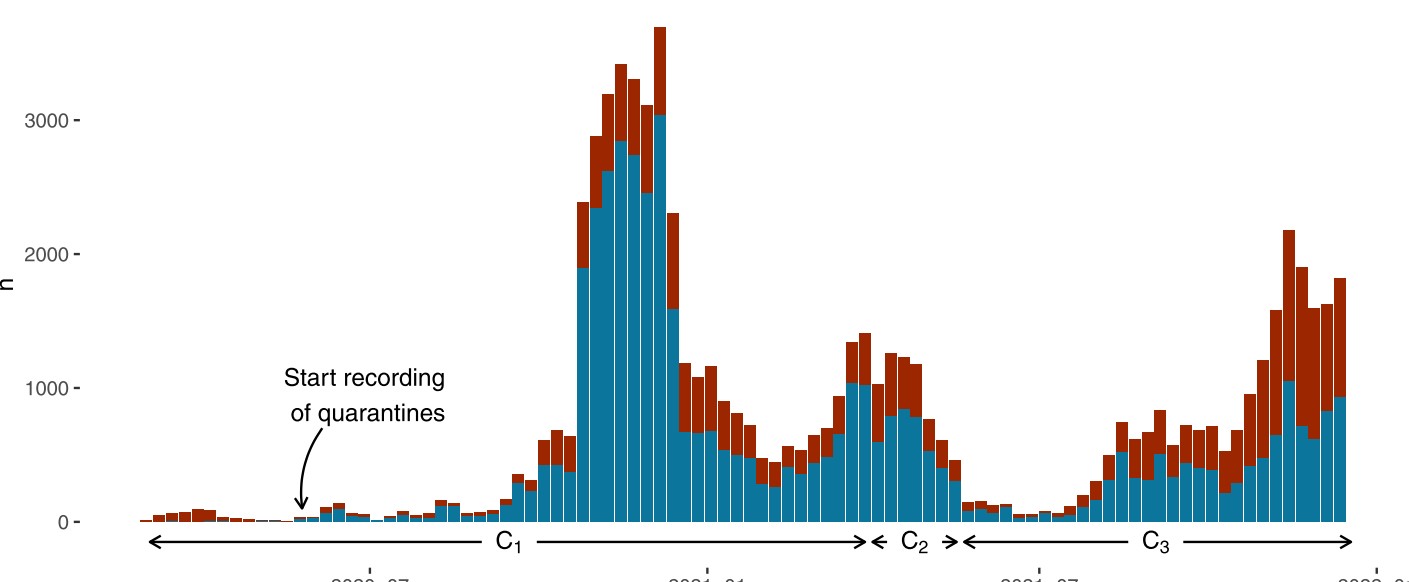

**Fig 1. Distribution of 24 433 COVID-19 isolation orders (in red) and 45 335 quarantine orders (in blue) over time in Berlin-Reinickendorf, Germany from March 2020 until December 2021.** Recommendation period for contact person definition: $C_1$ = 03 March 2020 to 30 March 2021, $C_2$ = 31 March 2021 to 19 May 2021, $C_3$ = 20 May 2021 to 18 December 2021. The distribution of isolation and quarantine orders over time shows the COVID-19 waves.

The median duration for isolation orders was $d_i$ = 10 days (interquartile range 8—13). The duration changed between different periods of recommendations: the median of the duration during the recommendation periods were: 14 days for $I_1$, 8 days for $I_2$ and 12 days for $I_3$. The overall median duration for quarantines was $d_q$ = 8 days (interquartile range 6—11). The median duration differed between periods of different recommendations and age groups: the median of the duration during the recommendation periods were 9 days for $Q_1$, 9 days for $Q_2$, 10 days for $Q_3$ and 4 days for $Q_4$. See Fig 2 for the distribution of age groups and recommended duration periods.

**Analysis of the ratio of contact persons per case.** In the time period after the 24 May 2020 (the first date, when contact persons were recorded in the software), the overall ratio of contact persons was $r_{qi}$ = 1.89. (2.88 in the contact person definition period no. 1, 1.96 in period no. 2 and 0.95 in period no. 3).

**Analysis of isolation orders following quarantine orders.** In the time period after the 24 May 2020, 3 483 of 45 272 quarantine orders had a directly following isolation order (contained case) and 535 had a following isolation order during the 1 to 7 days after the quarantine order (non-contained case). The 3483 contained cases represent 14% of the 24433 isolation orders. There was a difference between the different periods of recommendations see Fig 3. During the recommendation periods for the duration of quarantine $Q_{1-3}$, the percentage of non-contained cases was 1%; in $Q_4$ the percentage of non-contained cases was 3%.

**Analysis of timeliness.** There were two periods in which a duration of 14 days of quarantine was recommended ($Q_1$ and $Q_3$). During these periods, our approximation of the median time period between the last contact and the beginning of the quarantine order was $d_{delay}$ = 4 (interquartile range 1—6). The mean was 4.3 and 4 days for $Q_1$ and $Q_3$, respectively.

**Table 2. Analysis of 24 433 COVID-19 isolation and 45 335 quarantine orders by age and by period of contact person definition in Berlin-Reinickendorf, Germany from March 2020 to December 2021.**

| | N | $n_i$ | $n_q$ | $\tilde{d}_i$ | $\tilde{d}_q$ | $nd_{i-pi}$ | $nd_{q-pi}$ | $n_{cc}$ | $n_{cc}$ (%) | $n_{ncc}$ | $n_{ncc}$ (%) | $n_{ncc}/n_i$ | $n_i/n_q$ |
|---|---|---|---|---|---|---|---|---|---|---|---|---|---|
| total | 266123 | 24433 | 45335 | 10.2 | 8.3 | 0.9 | 1.4 | 3484 | 7.7% | 535 | 1.2% | 0.02 | 0.54 |
| 0—6 | 18084 | 1383 | 9149 | 11.2 | 8.2 | 0.9 | 4.1 | 434 | 4.7% | 97 | 1.1% | 0.07 | 0.15 |
| 7—17 | 27001 | 3983 | 17528 | 11.1 | 7.9 | 1.6 | 5.2 | 867 | 4.9% | 194 | 1.1% | 0.05 | 0.23 |
| 18—64 | 158199 | 16041 | 16678 | 10.1 | 8.7 | 1 | 0.9 | 1838 | 11% | 210 | 1.3% | 0.01 | 0.96 |
| 65—110 | 62839 | 3026 | 1980 | 9.4 | 8.4 | 0.5 | 0.3 | 345 | 17.4% | 34 | 1.7% | 0.01 | 1.5 |
| $C_1$ | 266123 | 10876 | 29973 | 8.3 | 8.9 | 0.3 | 1 | 1802 | 6% | 205 | 0.7% | 0.02 | 0.36 |
| $C_2$ | 266123 | 2446 | 4791 | 11.4 | 9.5 | 0.1 | 0.2 | 658 | 14% | 52 | 1.1% | 0.02 | 0.51 |
| $C_3$ | 266123 | 11111 | 10571 | 11.8 | 5.9 | 0.5 | 0.2 | 1024 | 10% | 278 | 2.6% | 0.03 | 1.05 |
| | N | $n_i$ | $n_q$ | $\tilde{d}_i$ | $\tilde{d}_q$ | $nd_{i-pi}$ | $nd_{q-pi}$ | $n_{cc}$ | $n_{cc}$ (%) | $n_{ncc}$ | $n_{ncc}$ (%) | $n_{ncc}/n_i$ | $n_i/n_q$ |
| total | 266123 | 24433 | 45335 | 10.2 | 8.3 | 0.9 | 1.4 | 3484 | 7.7% | 535 | 1.2% | 0.02 | 0.54 |
| 0—6 | 18084 | 1383 | 9149 | 11.2 | 8.2 | 0.9 | 4.1 | 434 | 4.7% | 97 | 1.1% | 0.07 | 0.15 |
| 7—17 | 27001 | 3983 | 17528 | 11.1 | 7.9 | 1.6 | 5.2 | 867 | 4.9% | 194 | 1.1% | 0.05 | 0.23 |
| 18—64 | 158199 | 16041 | 16678 | 10.1 | 8.7 | 1 | 0.9 | 1838 | 11% | 210 | 1.3% | 0.01 | 0.96 |
| 65—110 | 62839 | 3026 | 1980 | 9.4 | 8.4 | 0.5 | 0.3 | 345 | 17.4% | 34 | 1.7% | 0.01 | 1.5 |
| $C_1$ | 266123 | 10876 | 29973 | 8.3 | 8.9 | 0.3 | 1 | 1802 | 6% | 205 | 0.7% | 0.02 | 0.36 |
| $C_2$ | 266123 | 2446 | 4791 | 11.4 | 9.5 | 0.1 | 0.2 | 658 | 14% | 52 | 1.1% | 0.02 | 0.51 |
| $C_3$ | 266123 | 11111 | 10571 | 11.8 | 5.9 | 0.5 | 0.2 | 1024 | 10% | 278 | 2.6% | 0.03 | 1.05 |

N = number of inhabitants of Berlin-Reinickendorf in the specific group, $n_i$ = number of isolation orders, $n_q$ = number of quarantine orders, $\tilde{d}_i$ = mean duration of isolation orders, $\tilde{d}_q$ = mean duration of quarantine orders, $nd_{i-pi}$ = number of days in isolation per inhabitant, $nd_{q-pi}$ = number of days in quarantine per inhabitant, $n_{cc}$ = contained cases, quarantines that had a directly following isolation period, $n_{cc}$ (%) = $n_{cc}$ in percent of all quarantine orders, $n_{ncc}$ = non-contained cases, quarantine periods that had a following isolation period during day 1–7 after the quarantine period $n_{ncc}$ (%) = $n_{ncc}$ in percent of all quarantine orders. $C_{1-3}$ = period of contact person definition

## Discussion

Our analysis of the approximately 45 000 contact persons and 25 000 COVID-19 cases in Berlin-Reinickendorf quantifies the burden of isolation and quarantine orders for individuals infected with or exposed to COVID-19 with a total of 0.9 days of isolation and 1.4 days of quarantine per inhabitant. We showed that children were affected by quarantine orders to a larger extent than adults and that the local public health agency adapted the way of ordering separations when the Robert Koch Institute changed their recommendations. Our study found 3484 contained cases—15% of the total number of cases.

### Differences between children, adults and senior citizens

The local public health agency issued more quarantine orders per 100 inhabitants for children than for adults under 65 or for senior citizens. The same can be observed for isolation orders, but to a lesser degree, see Table 2. The Number of days in quarantine per inhabitant was roughly 20 times higher for school children than for senior citizens. The differences between the age groups can also be observed during different periods of recommendation. The higher number of isolation orders reflects the incidence of the age groups [27]. Our finding of the higher quarantine numbers could be due to the fact that children have more contacts than adults and even more than senior citizens (assortative contact patterns) [22]. Persons with a

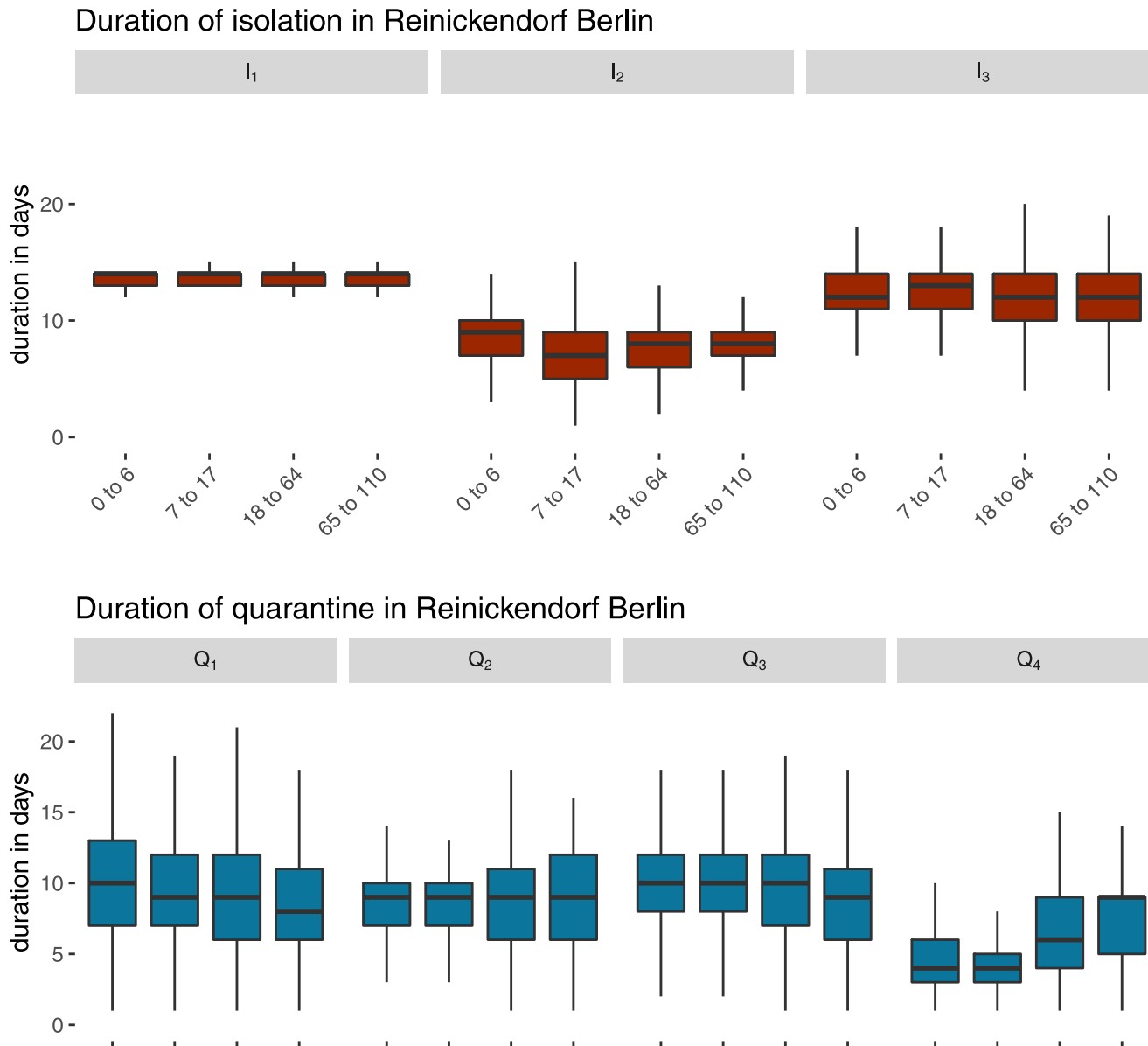

**Fig 2. Duration of isolation and quarantine orders by age group and recommendation period for COVID-19 between 3 March 2020 and 18 December 2021 in Berlin-Reinickendorf.** The recommended period of separation $Q_1$ from 3 March 2020 to 30 November 2020 was 14 days, in period $Q_2$ from 1 December 2020 to 15 February 2021 also 14 days, but allowed for testing at day 10, in period $Q_3$ from 16 February 2021 to 8 September 2021 again 14 days, in period $Q_4$ from 9 September 2021 to 18 December 2021 10 days, but allowed for testing at day 5 (children) or 7 (adults). The recommended period $I_1$ from 3 March 2020 to 1 July 2020 was 14 days, in period $I_2$ from 2 July 2020 to 30 March 2021 10 days, in period $I_3$ from 31 March 2021 to 18 December 2021 again 14 days.

higher number of contacts have a higher probability of being a contact person of a COVID-19 case. Thus, our differences in age groups might reflect only the higher number of contacts among children. Apart from a legitimate difference in quarantine orders per 100 inhabitants based on assortative contact patterns, our finding could also be due to the workflow of the

## Percentage of isolations that were preceded by a quarantine period

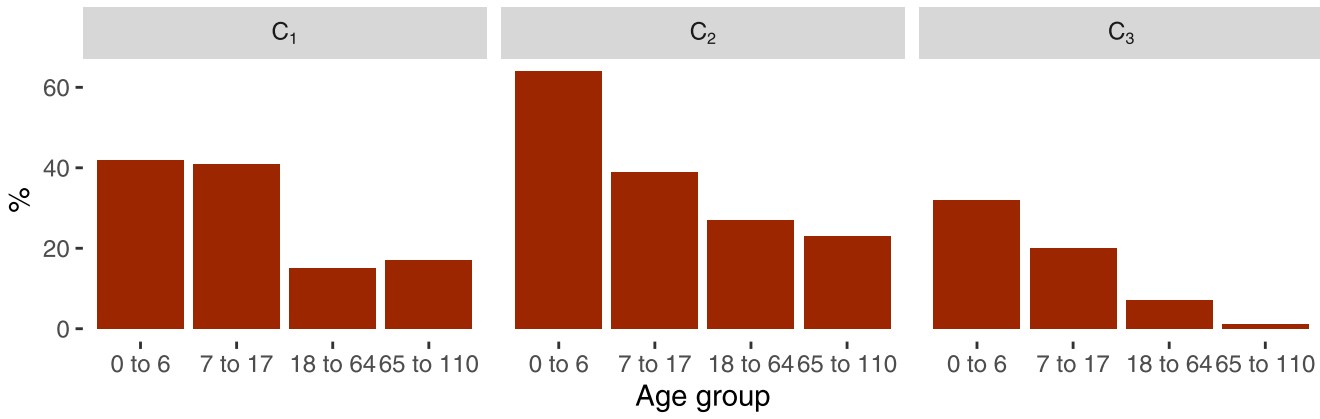

## Percentage of quarantines that were followed by an isolation period

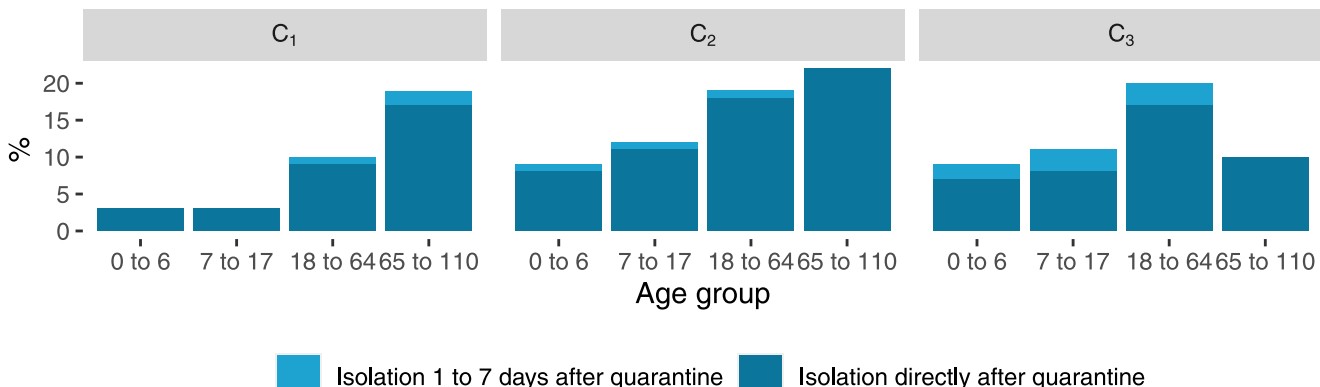

**Fig 3. Analysis of isolation orders following quarantine orders and vice versa.** Isolation orders that were preceded by a quarantine order in percent of all isolation orders and quarantine orders that were followed by an isolation order in percent of all quarantine orders. Analysis by age group and recommendation period of contact person definition ($C_1$ = 03 March 2020 to 30 March 2021, $C_2$ = 31 March 2021 to 19 May 2021, $C_3$ = 20 May 2021 to 18 December 2021.) between March 2020 and December 2021 in Berlin-Reinickendorf, Germany.

agency. The local public health agency needs to know about the contact and their address to order a quarantine. These preconditions are much more likely to be met amongst school and kindergarten children. It is easier to identify a child as a contact person than an adult. This would reinforce the feeling that children had to bear an unjustly high burden of the intervention, as it has been expressed widely throughout the media [28].

### Comparison of isolation and quarantine orders with school closures

Several studies, including large reviews, found that school closures reduce the transmission of the disease [29–31]. In the setting of Germany, Erhard et al. did not identify a significant increase of transmission after school openings in the state of Baden-Württemberg [32], whereas Sorg et al. calculated a decrease of 24% of expected cases during lockdown restrictions in Germany [33]. Compared to school closures, the intervention of isolation and quarantine orders was less costly in terms of school days lost. In Berlin, schools were closed for 124 days (from 13 March 2020 to 29 May 2020 and from 06 January 2021 to 22 February 2021). In this study, we found that the local public health agency ordered 6.8 days of separation order per schoolchild, which would be 1/18 of the number of school days lost from school closures.

However, for the individual affected by isolation and quarantine the individual separation order is probably perceived as a harsher measurement than school closure, because a school closure does not confine a person to their apartment. For future pandemics, decision makers have to carefully weigh the costs and benefits in terms of lost school and working days for each intervention.

## Average number of contact persons

Overall, the local public health agency identified less than two contacts per case. This is comparable with studies from the United States of America. Koetter et al. describes a student imitative that found 953 contacts for 536 cases which results in 1.8 contacts per case. Sachdev et al identified 0.7 contacts per case in the San Francisco Area. Shi et al. found 10.8 contacts per COVID-19 cases in China [17]. During the first outbreak in Germany 241 contacts from 17 cases were identified [34]. The mean number of contacts found in this study is much lower than the usual mean number of contacts per day (not contacts of COVID-19 cases) that Mossong et al. reported (7.95 mean contacts for Germany) [22]. The discrepancy between the usual number of contacts and the mean number of contacts found by us is even greater when it is taken into account that the contact definition used by the local health department includes aerosol contact, which would increase the average number of contacts compared to the study by Mossong et al. who studied face-to-face and skin-contact. During the pandemic, the contact monitor of the COVID-19 mobility project analysed telephone data and found 12.2 mean contacts per person (again not contacts of COVID-19 cases) and a decrease of the number of contacts at the beginning of the pandemic [35]. Our comparably low number of mean contacts can be considered to be a desired direct effect of the recommendation by the authorities to cut down contacts. But the identification of contacts is highly dependent on the workload of the agency and the rigour of contact tracing. So our low number of mean contacts per COVID-19 case could also indicate that not all contacts could be identified by the local public health agency or that cases were reluctant to give information about contacts.

## Effect of the contact tracing on the transmission of COVID-19

Contact tracing, which includes some sort of quarantine orders, has been implemented by 183 of 187 countries as measured by the Oxford COVID-19 Government Response Tracker [36]. It is considered to be one of the cornerstones of the response [37] and is also recommended by the WHO [1]. Pozo-Martin et al. found that there is evidence regarding the incremental effectiveness of both manual and digital contact tracing for COVID-19 epidemic control [38].

Isolation orders of infectious individuals and quarantine orders of contacts decrease the disease burden for the population, which has been shown in an epidemiological model by Agusto et al [39]. Nussbaumer-Streit et al. found a strong influence of quarantine (alone or in combination with isolation) on the reproductive number in a rapid review including 29 studies for the WHO [2]. On the other hand, empirical studies that compared several non-pharmaceutical interventions did not find an effect of quarantine on the spread of disease [29, 30]. The work of the local public health agency resulted in 3484 contained cases who were taken out of the transmission chain (persons that had an isolation period directly following a quarantine). We likely underestimated the number of contained cases, as our records included contact persons but not cases residing outside Berlin-Reinickendorf. Unlike contact persons, cases outside Berlin-Reinickendorf were managed by their respective local agencies, leading to potential underrepresentation in our data. Some of the contained cases might have spread the disease before they were ordered to quarantine (see the paragraph on timeliness) or during their quarantine order, because they did not adhere to the intervention.

## Influence of contact person definition

Our data show that during times with a sensitive contact person definition (Contact person definition period $C_1$) the mean number of contacts per COVID-19 case was higher than during times with a less sensitive contact person definition (Contact person definition period $C_{2-3}$). Consequently, during times with a sensitive definition, the percentage of contained cases was lower and the number of cases that were previously identified as contact persons was higher. This is consistent across all age groups as illustrated in Fig 3. It seems plausible that a change in the recommended definition results in a different number of identified contact persons, however our evidence must be considered as weak, see limitation section. The right balance between specificity and sensitivity for the contact person definition depends on the strategy of the government.

## Effect of the recommendations of the Robert Koch Institute on duration of isolation and quarantine orders

The results show that the duration of isolation and quarantine orders changes along with the recommended period, which indicates that the local public health agency adhered to the recommendations by the Robert Koch Institute. Fig 2 shows a clear pattern of the duration that correlates with the recommendations as given in Table 1. This result indicates that the recommendations caused the change in duration of isolation and quarantine orders.

## Timeliness of contact tracing

The calculated median duration of quarantine orders was lower than the recommended time by the Robert Koch Institute which is due to the fact that there is a delay between the date of contact and the identification and subsequent quarantine order by the local public health agency. The following steps take place between the contact to the index case and the quarantine order: The index case conducts a test, the test result is reported and processed, the agency reaches out to the index case to identify the contact persons and contacts them. The median delay of approximately 4 days in contact tracing, as identified in our study, represents a shortcoming in the containment efforts for COVID-19. Prompt detection of contact persons is crucial to effectively prevent the transmission of the virus [40].

## Effect of the duration of quarantines on the number of contained cases

During the course of the pandemic, the recommended time period for the isolation and quarantine period was changed several times. A reduction in the duration of quarantine increases the risk of having non contained cases (contact persons that turned into cases not directly but one to seven days after their quarantine order). Chinese authors suggest a quarantine duration for longer than 14 days—which was the maximum in Germany—based on a calculated 9% of total cases that had symptoms or other events beyond 14 days [41]. In Europe, Ashcroft et al. suggested that a reduction of the quarantine period from 10 to 7 days combined with testing strategies can be a feasible method to reduce the burden of quarantine for contacts and returning travellers [42]. The recommendations of the Robert Koch Institute on the duration of quarantine ($Q_4$) proposed a reduction of the quarantine duration similar to what Ashcroft et al. suggested—the main exception being a possibility for children to end the quarantine after 5 days with a negative test. Comparing the time period for quarantine duration before and after this change ($Q_{1-3}$ vs. $Q_4$), we found an increase in the number of non contained cases from 1% to 3%. For the measurement of this analysis our data is limited—see limitation section.

### Limitations of this work

Our data has several limitations. Firstly, the primary aim of the local public health agency is not to acquire data for for scientific purposes but to prevent the spread of the disease. Thus, data was not collected with the scientific scrutiny. Secondly, the data correctness depends on the number and experience of the staff. The staff had a high turnover during the pandemic and the workload changed several times during the pandemic. Other factors, however, favour the correctness of data collection: the local public health agency is required by law to do contact tracing [43], persons usually need a document stating that they are placed in isolation or quarantine—to generate this document the isolation or quarantine order needs to be entered into the reporting software. We made efforts to minimise the error due to work overload by reducing our analysis to the time period without excessive work overload. Thirdly, there are severe limitations in the time dependent analysis, e.g., the evaluation of the effect of different recommendations, since we are not able to disentangle the different effects on our measured variables due to the many number of parameters that changed during the course of the pandemic. Besides the different recommendations on isolation and quarantine duration and the definition of contact persons, confounding variables include: vaccination status, variants of the virus, work load, number of staff, experience of staff, different contact pattern of cases, testing behaviour, contact person behaviour. Fourth, for the analysis that involved a comparison of isolation and quarantine orders, we could not directly link a case to a contact person (thus we could not calculate secondary attack rates by type of contact or other useful measures like direct number of contacts per person). Fifth, contact persons that turned into a case were sent to another local public health agency if they resided outside of Berlin-Reinickendorf. A sixth limitation is that our data reflects only the direct work of the local public health agency and cannot measure the indirect effects: inhabitants of Berlin-Reinickendorf may have been isolating or quarantining themselves or on the basis of the law of the federal state or the district decree without giving a notification to the local public health agency. Seventh, the true number of cases is not known and so impact of the public health agency is underestimated.

### Conclusion

Isolation of COVID-19 infected individuals and quarantine of contacts is one important tool to slow down the pandemic. However, separation orders cause health hazards, such as mental health impacts. Our study concludes the following for Berlin-Reinickendorf: the local public health agency ordered 1.4 days of quarantine and 0.9 days of isolation per inhabitant. The local public health agency contained 3484 cases. Contact tracing places a burden on the population, but the number of days lost due to isolation or quarantine are much fewer than the days lost to school closures or work closures. The local public health agency found 1.9 contacts per case— clearly lower than Chinese agencies or the investigations during the first outbreak in Germany or the usual mean number of contacts (in the absence of COVID-19). This indicates that the agency was not able to provide rigorous contact tracing. Children were quarantined to a much higher degree than adults or senior citizens. Our data indicate that the recommendations by the Robert Koch Institute had an influence on the work of the local public health agency. With limitations, we found a delay of 4 days between the date of contact and the date the contact person was ordered to quarantine.

### Supporting information

**S1 Table. Detailed table of time periods of relevant recommendations of the Robert Koch Institute for isolation duration, quarantine duration and contact person definition.**
(PDF)

**S1 Fig. Exclusion of data entries.**
(PDF)

## Acknowledgments

We thank Patrick Larscheid for his organisational support and the staff of the local public health agency for their work in protecting the people of Berlin-Reinickendorf. Many thanks to Maria Helmrich for her help with the wording.

## Author Contributions

**Conceptualization:** Jakob Schumacher.

**Data curation:** Jakob Schumacher.

**Formal analysis:** Jakob Schumacher, Sonja Jäckle.

**Investigation:** Lisa Kühne, Sophia Brüssermann.

**Methodology:** Jakob Schumacher, Lisa Kühne, Sophia Brüssermann, Benjamin Geisler, Sonja Jäckle.

**Software:** Sonja Jäckle.

**Validation:** Sonja Jäckle.

**Writing – original draft:** Jakob Schumacher, Lisa Kühne, Sophia Brüssermann, Benjamin Geisler, Sonja Jäckle.

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
