## [Decision Letter · Decision Letter 0]

19 Aug 2022

PONE-D-22-19240COVID-19 isolation and quarantine orders in Berlin-Reinickendorf (Germany): How many, how long and to whom?PLOS ONE

Dear Dr. Jakob Schumacher,

Thank you for submitting your manuscript to PLOS ONE. After careful consideration, we feel that it has merit but does not fully meet PLOS ONE’s publication criteria as it currently stands. Therefore, we invite you to submit a revised version of the manuscript that addresses the points raised during the review process.

The points need to addressed and incorporated in the revised manuscript.

We look forward to receiving your revised manuscript.

Kind regards,

Srikanth Umakanthan

Academic Editor

PLOS ONE

Journal Requirements:

Additional Editor Comments:

Kindly address all the points as mentioned in the recommended revision comments.

Reviewers' comments:

Reviewer's Responses to Questions

**Comments to the Author**

1. Is the manuscript technically sound, and do the data support the conclusions?

Reviewer #1: Yes

Reviewer #2: Yes

2. Has the statistical analysis been performed appropriately and rigorously? 

Reviewer #1: Yes

Reviewer #2: Yes

3. Have the authors made all data underlying the findings in their manuscript fully available?

Reviewer #1: Yes

Reviewer #2: Yes

4. Is the manuscript presented in an intelligible fashion and written in standard English?

Reviewer #1: Yes

Reviewer #2: Yes

5. Review Comments to the Author

Reviewer #1: The authors would benefit by including the following suggestions to strengthen the manuscript:

1. Introduction on COVID-19 with emphasis on its origin and transmission (refer and cite: doi: 10.1136/postgradmedj-2020-138234).

2. Include the role of vaccination (refer and cite: doi: 10.3390/vaccines91010640

3. Include the role of protective medications in individuals with high risk of COVID-19 (refer and cite: doi: 10.1186/s41231-021-00102-4).

4. Include a comparison of support programs in Germany with that of other regions (refer and cite: doi: 10.3389/fpubh.2022.844333)

5. Role of predictors for vaccine hesitancy in Germany. (refer and cite: doi: 10.1136/postgradmedj-2021-141365.

6. where there any inclusion and exclusion criteria among the selected populations?

7. Where there any missing records? Kindly mention in your methodology.

8. Mention if any limitations were identified in your study and also mention the follow up steps for further analysis.

Reviewer #2: Well written manuscript and I endorse it for publication. The authors have well versed the materials, methods and statistical analysis. The results are well described and illustrated in the figures and tables.

6. PLOS authors have the option to publish the peer review history of their article (what does this mean?). If published, this will include your full peer review and any attached files.

Reviewer #1: No

Reviewer #2: No

---

## [Author Response · Author response to Decision Letter 0]

2 Nov 2022

Dear ladies and gentlemen,

thank you for your review of the manuscript. Please find our changes in the manuscript. I hope that I corrected all you points and got the latex and naming conventions right. 

Best regards

Jakob Schumacher and the coauthors

Points raised by Reviewer No. 1

1. Introduction on COVID-19 with emphasis on its origin and transmission (refer and cite: doi: 10.1136/postgradmedj-2020-138234).

2. Include the role of vaccination (refer and cite: doi: 10.3390/vaccines91010640)

3. Include the role of protective medications in individuals with high risk of COVID-19 (refer and cite: doi: 10.1186/s41231-021-00102-4).

4. Include a comparison of support programs in Germany with that of other regions (refer and cite: doi: 10.3389/fpubh.2022.844333)

5. Role of predictors for vaccine hesitancy in Germany. (refer and cite: doi:10.1136/postgradmedj-2021-141365.

6. where there any inclusion and exclusion criteria among the selected populations?

7. Where there any missing records? Kindly mention in your methodology.

8. Mention if any limitations were identified in your study and also mention the follow up steps for further analysis.

Answers to Reviewer No. 1

Thank you for your contributions, please find our responses to your points beneath

Answer to point 1-3 and 5

We included a short introduction to COVID-19 including its origin, pathogenesis and transmission

as well as the role of vaccination, vaccine hesitancy and protective medications. We tried to keep it

short to not waste the time of the reader. Please see the revised manuscript.

1. Holmes EC, Goldstein SA, Rasmussen AL, Robertson DL, Crits-Christoph A, Wertheim JO,

et al. The origins of SARS-CoV-2: A critical review. Cell. 2021;184(19):4848–4856.

doi:10.1016/j.cell.2021.08.017.

2. Hu B, Guo H, Zhou P, Shi ZL. Characteristics of SARS-CoV-2 and COVID-19. Nature

Reviews Microbiology. 2021;19(3):141–154. doi:10.1038/s41579-020-00459-7.

3. Fiolet T, Kherabi Y, MacDonald CJ, Ghosn J, Peiffer-Smadja N. Comparing COVID-19

vaccines for their characteristics, efficacy and effectiveness against SARS-CoV-2 and variants

of concern: a narrative review. Clinical Microbiology and Infection. 2022;28(2):202–221.

doi:10.1016/j.cmi.2021.10.005.

4. Pires C., Global Predictors of COVID-19 Vaccine Hesitancy: A Systematic Review. Vaccines.

2022;10(8):1349. doi:10.3390/vaccines10081349.

Answer to point 4

Support programs are an important part in the response to COIVD-19. We feel that the

comparison of support programs with different regions would merit its own publication. We decided

to not include such a comparison to not extend the scope of our publication.

Answer to point 6 and point 7

Thank you for your point. We have not been clear enough in displaying our population, which we

addressed in our revised manuscript. In our opinion we have already addressed inclusion and

exclusion criteria as well as our handling of missings by stating that we took all the entries in our

database (from the study period) on separation orders. Then we described which of those orders we

took (all with a valid person ID with no missing data in the variables ”separation from” and

”separation until”. We then proceeded to remove all entries that we regarded as a typing error and

then combining overlapping order double entries of separation orders. However we might not have

been clear enough in our description so we addressed that point to make it more understandable.

Answer to point 8

Agreed, limitations are an important part in every publication. We have identified five limitations

that we feel important: 1) Aim of data collection, 2) Overload of work that alters data collection 3)

changes over time 4) No direct linking of cases and their contact persons and 5) indirect effects. We

also wrote what we think limits the influence of these limitations. We included a paragraph on

those five limitations. Also when we feel that on of our conclusions had a limitation that addressed

it directly we tried to make that clear directly beside that point. We revised the section on

limitations to make it more understandable and to better show their importance.

Points raised by Reviewer No. 2 

The authors have well versed the materials, methods and statistical analysis. The results are well described and illustrated in the figures and tables. We included the following additional citations

Answers to Reviewer No. 2

Thank you for you positive feedback

---

## [Editor Report · Decision Letter 1]

6 Jan 2023

PONE-D-22-19240R1COVID-19 isolation and quarantine orders in Berlin-Reinickendorf (Germany): How many, how long and to whom?PLOS ONE

Dear Dr. Jakob,

Thank you for submitting your manuscript to PLOS ONE. After careful consideration, we feel that it has merit but does not fully meet PLOS ONE’s publication criteria as it currently stands. Therefore, we invite you to submit a revised version of the manuscript that addresses the points raised during the review process.

Kindly incorporate all the points mentioned by the reviewers in the revised manuscript.

We look forward to receiving your revised manuscript.

Kind regards,

Srikanth Umakanthan

Academic Editor

PLOS ONE

Journal Requirements:

Additional Editor Comments (if provided):

The authors have not incorporated the changes mentioned by the reviewers. Minor revision still stands for the submitted manuscript.

---

## [Author Response · Author response to Decision Letter 1]

24 Feb 2023

Please find the latest E-Mail regarding the Reviewprocess with the latest E-Mail being:

Dear Dr. Schumacher,

Apologies for the delay in getting back to you. I've consulted with our internal team and want to let you know that you do not need to cite the papers requested by Reviewer 1. You are welcome to resubmit your manuscript at any time.

We will follow up internally regarding your concerns about this review. If you have any further questions, please feel free to reach out.

Kind regards,

Sabine Henderson

---

## [Decision Letter · Decision Letter 2]

5 Dec 2023

PONE-D-22-19240R2COVID-19 isolation and quarantine orders in Berlin-Reinickendorf (Germany): How many, how long and to whom?PLOS ONE

Dear Dr. Schumacher,

Thank you for submitting your manuscript to PLOS ONE. After careful consideration, we feel that it has merit but does not fully meet PLOS ONE’s publication criteria as it currently stands. Therefore, we invite you to submit a revised version of the manuscript that addresses the points raised during the review process.

We look forward to receiving your revised manuscript.

Kind regards,

Emanuele Crisostomi, PhD

Academic Editor

PLOS ONE

Journal Requirements:

**Additional Editor Comments:**

This Academic Editor received the assignment of this manuscript only three weeks ago, and realized that the manuscript had a past long history with unsuccessful attempts to finding appropriate reviewers, since the original ones were not available for a second round of reviews.

The new reviews are overall positive, and a minor review is recommended. In particular, one Reviewer provides a number of punctual constructive comments for improving the manuscript.

The next round of reviews will be faster and a final decision will be taken.

Reviewers' comments:

Reviewer's Responses to Questions

**Comments to the Author**

1. If the authors have adequately addressed your comments raised in a previous round of review and you feel that this manuscript is now acceptable for publication, you may indicate that here to bypass the “Comments to the Author” section, enter your conflict of interest statement in the “Confidential to Editor” section, and submit your "Accept" recommendation.

Reviewer #3: (No Response)

Reviewer #4: All comments have been addressed

Reviewer #5: (No Response)

2. Is the manuscript technically sound, and do the data support the conclusions?

Reviewer #3: Partly

Reviewer #4: Yes

Reviewer #5: Yes

3. Has the statistical analysis been performed appropriately and rigorously? 

Reviewer #3: Yes

Reviewer #4: Yes

Reviewer #5: Yes

4. Have the authors made all data underlying the findings in their manuscript fully available?

Reviewer #3: (No Response)

Reviewer #4: Yes

Reviewer #5: (No Response)

5. Is the manuscript presented in an intelligible fashion and written in standard English?

Reviewer #3: Yes

Reviewer #4: Yes

Reviewer #5: Yes

6. Review Comments to the Author

Reviewer #3: Overall this is an interesting account of the work of a health department during the COVID-19 pandemic, including counts of isolation and quarantine orders as well as intervals e.g. between contact and beginning of quarantine. A few amendments might make the paper more informative. Reading the manuscript by a native speaker may be advisable.

Abstract:

-The intro could be strengthened (the harm infliction is not a great part of the paper, is it?), and please add what the aim of the paper is.

-not sure if the analysis on the “dependence on the recommendations by the Robert Koch Institute” is that informative for the readers, I suggest to de-emphasize this part throughout.

-“delay of 4 days between contact and quarantine order”: what is meant: “last contact”? what did you do with family members as there was no point contact, but usually continuous contact? (perhaps address in Methods (were they excluded for that purpose?))

-“3484 of contacts were in quarantine…”: relate to the number of cases (1/18) and to the number of quarantines, and eliminate the word “of”. If you are able to quantify the number of human resources (man-hours or something like that) you could relate to that measure, too.

-I suggest to add here the proportion of the population that received at least one isolation or quarantine order, for simplicity sake you could assume that all isolation and quarantine orders were different persons

-the claim that fewer days were lost due to isolation or quarantine orders is not quantified.

-I would change the conclusions. (1)the part with the Robert KochInstitute I would leave out, (2)the conclusion that the mean number of contacts is less than expected comes out of the blue as you have not said what you expected. For example: you could say that a substantial proportion of the population had been separated, that the efforts saved the children from school closures, and that a substantial number of cases occurred when in quarantine.

Methods

-perhaps start with how you did the routine work and the contact tracing, how you monitored quarantined persons if they became a case (self monitoring / health department monitoring, symptom monitoring, mandatory testing?, …)

-line 73: what was the unit of observation in that database? an order for isolation, or an order to quarantine? either?

-what you could do and would be quite cool is to quantify the cases by the day they were removed from the public, e.g. x1 number of cases on day -2 before symptom onset (because they were in quarantine), x2 number of cases from day -1 before symptom onset, x3 number of cases on the day when they had symptom onset, etc. I think that should be feasible and would be quite good information for modelers.

-paragraph on “Number of quarantine orders…”: please give examples. Line 130: “other cases”: not clear what is meant here.

-please explain here how you dealt with household contacts from the point of view “last contact”. Did you exclude household contacts because they usually had no fixed, identifiable single contact?

Results

-Line 149: there are two “??”, please complete manuscript

-Fig.1: Since this is a paper that gives detailed account of the efforts of a health department I suggest to overlay the number of persons that were involved with contact tracing (by week or month, if possible) to show the amount of work and resources that has gone into this effort.

- can you give also the number of cases reported, please?, perhaps also in Table 2 and/or Fig.1

- please describe / point out the most important things in the figure/table (don’t leave the reader alone with it)

- Table 2, footnotes: add what N means

-Line 192: the number of contacts per case is quite small. given that these include also cases in schools with likely substantially more contacts it suggests that mainly the household contacts were traced. Please add in the Limitations that this low number of contacts traced may be a reason that the percentage of contained cases of all isolation orders is not more than 14%. At any rate: could you please give a break-down of the type of contacts, e.g. household, work, school, other, or something like that? In fact, I would consider restricting the analysis of contacts to household contacts as I assume that the vast majority of contacts are household contacts. This would strengthen the paper.

- could you add in the table the ratio of contained cases per quarantined persons (ncc/nq) or ni/nq (isolations per quarantined persons)

- I think the ratio of isolations in the 1-7 days after end of quarantine : isolations during quarantine is a useful measure that should be provided.

- Lines 199/200: what do you mean by “non-contained cases”? please give raw numbers here, too.

-perhaps provide also the percentage of the population with isolation order or quarantine order?

Discussion

-Limitations are usually at the end of the paper unless the editors wish something else

-I think another limitation is that there are also cases that were never known to the health department, which is of course not the fault of the health department, however, since you do not know the true number of cases (including those never diagnosed who still contribute to transmission) you underestimate the impact of the health department

-Line 232: does that mean you could not calculate secondary attack rates, e.g. by type of contact (household, work, etc.) or age of contact, or an average number of cases by source case (e.g. to identify super-spreaders)?

-Line 240: I cannot extract that or at least not easily from Table 2, please be more precise?

-Line 283: … or a reluctance of cases to give information about contacts

-Line 295: an update has been published, however, a lot of the evidence stems from SARS and MERS; there is other publications with a focus on C-19, such as Pozo-Martin (EJE, 2023, "Comparative effectiveness..."), Craig (JMIR 2021, "Effectiveness of contact tracing...") and Fetzer (PNAS, 2021, "Measuring the scientific effectiveness...")

-Line 299: I dont understand what is meant here

-Line 327: “The rough estimate for the median delay of 4 days that we found must be considered as a flaw in the contact tracing. For COVID-19, an early detection of contact persons is key to hinder transmission of the disease” – please check English (particularly “flaw” and “hinder” ??)

-but did adherence increase and/or administrative feasibility improve?

-Line 357: for modellers that is good information. however, again, most informative would be the proportion of cases that were isolated by day in relation to symptom onset (see above).

- The authors calculated the number of contacts per case. I think it would be more instructive to stratify by those cases who were not previously in the system as contact person and those who were. The latter will likely have very few contacts, the first might have a number of contacts. In addition it would be nice to stratify by age (of the case; very broadly, say <18 and 18+ year old), because children should have quite a bit more contacts than adults.

Reviewer #4: This is a nicely conducted and well-written paper that provides detailed descriptives on the COVID-19 isolation and quarantine orders by age groups and time periods in the Berlin-Reinickendorf area in Germany. Overall, I think the manuscript is well-structured and the analyses are appropriate in answering the research questions. The authors have also responded to the previous review comments sufficiently.

I only have few minor comments for consideration before publication:

- It will help the readers understand the context better if the authors can also state in the Abstract the total number of individuals analyzed.

- In Table 2, it will be helpful to mark the percentages as “%”, and provide SD for mean values. The abbreviations are also a bit hard to understand at first glance (for example it is not intuitive to me how “ncc” refers to “quarantines that had a directly following isolation period”). As these abbreviations are used only once in Table 2, the authors can consider write them out in words or improve the labels, if possible.

- It is generally recommended to avoid using the term “elderly”. I suggest changing it to “older adults”, “older persons”, or similar throughout the manuscript.

Reviewer #5: This is an intriguing study that provides valuable insights into policies regarding isolation and quarantine for COVID-19. The methodology is robust, and the contents are well-written. The results not only capture a specific time during the COVID-19 era but also offer potential implications for future pandemics or emerging diseases. I have only a few minor comments:

1. I noticed that reference 21 was not cited in the manuscript. Additionally, please renumber the references in the order they appear in the manuscript. As an example, references 36-38 currently appear in the first paragraph of the Introduction, which requires revision for proper numbering.

2. Line 11-12: "Vaccines have been developed and proven successful in preventing severe disease and death [40], ...." → I suggest change this sentence to "Vaccines have been developed and proven successful in inducing seroconversion (cite the following reference 1), preventing severe disease and death [40], ...."

Reference:

[1] Safety and Seroconversion of Immunotherapies against SARS-CoV-2 Infection: A Systematic Review and Meta-Analysis of Clinical Trials. Pathogens. 2021 Nov 24;10(12):1537. doi: 10.3390/pathogens10121537. PMID: 34959492; PMCID: PMC8706687.

3. Line 51-52: "Jian et al. reported on the Taiwanese digital system TRACE analysing 487 cases and 8051 contact persons [12]." → To emphasize the public health implications of these particular systems, I suggest change this sentence to: "Jian et al. reported on the Taiwanese digital system TRACE, analyzing 487 cases and 8051 contact persons [12]. Public health implications of these systems included surveillance of travel history (cite the following reference 1) and ensuring an adequate quantity of personal protective equipment (cite the following reference 2)."

References:

[1] Integrating travel history via big data analytics under universal healthcare framework for disease control and prevention in the COVID-19 pandemic. J Clin Epidemiol. 2021 Feb;130:147-148. doi: 10.1016/j.jclinepi.2020.08.016

[2] Big Data-driven personal protective equipment stockpiling framework under Universal Healthcare for Disease Control and Prevention in the COVID-19 Era. Int J Surg. 2020 Jul;79:290-291. doi: 10.1016/j.ijsu.2020.05.091

I look forward to reviewing a revised version of this work based on my feedback!

7. PLOS authors have the option to publish the peer review history of their article (what does this mean?). If published, this will include your full peer review and any attached files.

Reviewer #3: No

Reviewer #4: No

Reviewer #5: No

---

## [Author Response · Author response to Decision Letter 2]

21 Dec 2023

Points raised by Reviewer No. 3

Thank you for your thorough feedback and the time you spend on improving the

paper.

1. The intro could be strengthened (the harm infliction is not a great part of the paper, is it?),

and please add what the aim of the paper is.

The introduction aims to provide a comprehensive overview, including potential

negative impacts of isolation and quarantine measures. The publication is not

about what adverse effects are done by quarantine and isolation but to measure

the quantity of isolations and quarantines. In my eyes the aim is quite cleary

covered with: The goal of this work is to quantify and analyse isolation and

quarantine orders from a local public health agency in Germany to assess

differences among age groups, mean number of contacts, timeliness of quarantine

orders, the number of contained or non-contained cases and to estimate the

impact of the contact person tracing. I updated the intro to improve the part

about the adverse effects.

2. not sure if the analysis on the “dependence on the recommendations by the Robert Koch

Institute” is that informative for the readers, I suggest to deemphasize this part throughout.

Thank you for your input regarding the analysis on the dependence on the

Robert Koch Institute’s recommendations. I believe this aspect is crucial as it

highlights the significance of these recommendations in guiding public health

responses. I think it is important to know for future pandemics if local agencies

follow or dont follow national recommendations. This analysis reflects the impact

of these guidelines and shows that the national institute plays a key role.

Therefore, I suggest not to deemphasize this part.

3. “delay of 4 days between contact and quarantine order”: what is meant: “last contact”? what

did you do with family members as there was no point contact, but usually continuous

contact? (perhaps address in Methods (were they excluded for that purpose?))

Thank you for your point regarding the ’delay of 4 days between contact and

quarantine order’ and the definition of ’last contact’, especially in the context of

family members. To clarify, I have added a paragraph in the Methods section

explaining ’last contact’ in the paragraph explaining the timeliness.

4. “3484 of contacts were in quarantine. . . ”: relate to the number of cases (1/18) and to the

number of quarantines, and eliminate the word “of”. If you are able to quantify the number of

human resources (man-hours or something like that) you could relate to that measure, too.

Point taken. I changed the sentence in the introduction to relate to the number

of cases and quarantines. Unfortunately we were not able to quantify the number

of human resources, although I agree that this would be a valuable addition

5. I suggest to add here the proportion of the population that received at least one isolation or

quarantine order, for simplicity sake you could assume that all isolation and quarantine orders

were different persons

Thank you for your point. I added the proportion of the population as you

suggested.

6. the claim that fewer days were lost due to isolation or quarantine orders is not quantified.

I would argue that I did quantify this aspect with the sentence in the discussion:

Compared to school closures, the intervention of isolation and quarantine orders

was less costly in terms of school days lost. In Berlin, schools were closed for

124 days (from 13 March 2020 to 29 May 2020 and from 06 January 2021 to 22

February 2021). In this study, we found that the local public health agency

ordered 6.8 days of separation order per schoolchild, which would be 1/18 of the

number of school days lost from school closures. To clarify I moved bits of this

into the introduction, which is probably the better position for this sentence.

7. I would change the conclusions. (1)the part with the Robert KochInstitute I would leave out,

(2)the conclusion that the mean number of contacts is less than expected comes out of the

blue as you have not said what you expected. For example: you could say that a substantial

proportion of the population had been separated, that the efforts saved the children from

school closures, and that a substantial number of cases occurred when in quarantine.

As stated above I think the part with the recommendations of the RKI is

important, so I would suggest to leave it in. In the paper we compared our

findings to the literature that calculates around 8 contacts per day. I updated the

abstract and the introduction to better clarify that point. I included your

suggestions for further conclusions.

8. Methods: perhaps start with how you did the routine work and the contact tracing, how you

monitored quarantined persons if they became a case (self monitoring / health department

monitoring, symptom monitoring, mandatory testing?, . . . )

Agreed, this information is missing. I added a paragraph that includes this.

9. line 73: what was the unit of observation in that database? an order for isolation, or an order

to quarantine? either?

That was indeed unclear. I clarified that I meant both.

10. what you could do and would be quite cool is to quantify the cases by the day they were

removed from the public, e.g. x1 number of cases on day -2 before symptom onset (because

they were in quarantine), x2 number of cases from day -1 before symptom onset, x3 number

of cases on the day when they had symptom onset, etc. I think that should be feasible and

would be quite good information for modelers.

Thank you for your interesting suggestion. While this approach should indeed

offer valuable insights for epidemiological modelling the information on the onset

of disease is not available in the current data.

11. paragraph on “Number of quarantine orders. . . ”: please give examples. Line 130: “other

cases”: not clear what is meant here.

I have revised the paragraph with the aim of enhancing its readability.

12. please explain here how you dealt with household contacts from the point of view “last

contact”. Did you exclude household contacts because they usually had no fixed, identifiable

single contact?

Thank you for your point. I added a sentence to explain our management in

household situations

13. Line 149: there are two “??”, please complete manuscript

There was a latex-problem that is hopefully solved now

14. Fig.1: Since this is a paper that gives detailed account of the efforts of a health department I

suggest to overlay the number of persons that were involved with contact tracing (by week or

month, if possible) to show the amount of work and resources that has gone into this effort.

That would indeed be great if we had the numbers on that. Unfortunately we did

not track the number of persons involved with contact tracing sufficiently

15. can you give also the number of cases reported, please?, perhaps also in Table 2 and or Fig 1

Added this missing information

16. please describe point out the most important things in the figure/table (don’t leave the reader

alone with it)

We previously had a discussion in the team whether to show this graph because it

does not directly lead to any conclusions. In the end we felt that it has merit to

give the reader a sense about the distribution of the cases and contacts over time.

I added a sentence to include this information.

17. Table 2, footnotes: add what N means

This was indeed missing

18. Line 192: the number of contacts per case is quite small. given that these include also cases in

schools with likely substantially more contacts it suggests that mainly the household contacts

were traced. Please add in the Limitations that this low number of contacts traced may be a

reason that the percentage of contained cases of all isolation orders is not more than 14%. At

any rate: could you please give a break-down of the type of contacts, e.g. household, work,

school, other, or something like that? In fact, I would consider restricting the analysis of

contacts to household contacts as I assume that the vast majority of contacts are household

contacts. This would strengthen the paper.

For me personally this is one of the most important findings of our study. We

expect to have usually around 8 contacts per person per day (Mossong et al.) and

for the contact definition we were looking for up to 14 days of potential contacts.

So we should have a lot more than 2 contacts per case. Although one could argue

that people cut down their contacts during the pandemic. Chinese papers

reported around 10 contacts per case which is maybe why they managed to

reduce the number of cases close to zero. So if we have another pandemic where

we want to follow a zero-strategy we should consider stepping up the search for

contacts. The other point - breaking down by type of contact - would be a

valuable addition. But this information was not recorded in the software. You

can see in the later parts of the publication the breakdown by age which gives an

indication for the difference between schools and household. Interestingly the

percentage of contained cases goes up if you have lower numbers of contacts per

case - probably because you get the most important contacts only. But the

absolute number of contained cases would rise the more contacts are found.

19. could you add in the table the ratio of contained cases per quarantined persons (ncc/nq) or

ni/nq (isolations per quarantined persons)

The ratio of contained cases per quarantined persons is already present as a

percentage in the colum ncc-p. I added the % symbol to make it clearer. I added

the ni/nq (isolations per quarantined persons).

20. I think the ratio of isolations in the 1-7 days after end of quarantine : isolations during

quarantine is a useful measure that should be provided.

I added the ratio in the column nncc/ni

21. Lines 199/200: what do you mean by “non-contained cases”? please give raw numbers here,

too.

What we mean with non-contained case is explained in the methods and in the

mentioned paragraph two sentences earlier. We wanted to avoid writing had a

following isolation order during the 1 to 7 days after the quarantine order several

time.

22. perhaps provide also the percentage of the population with isolation order or quarantine

order?

We have: The local public health agency ordered ni = 24 433 isolations and nq =

45 335 quarantines (ni-p100 = 9.2 isolation orders and nq-p100 = 17 quarantine

orders per 100 inhabitants). We hope that this gives adequate information.

Discussion

23. Limitations are usually at the end of the paper unless the editors wish something else

Thanks for the clarification: I moved it to the end

24. I think another limitation is that there are also cases that were never known to the health

department, which is of course not the fault of the health department, however, since you do

not know the true number of cases (including those never diagnosed who still contribute to

transmission) you underestimate the impact of the health department

I added the limitation.

25. Line 232: does that mean you could not calculate secondary attack rates, e.g. by type of

contact (household, work, etc.) or age of contact, or an average number of cases by source

case (e.g. to identify super-spreaders)?

Yes, we would have loved to calculate secondary attack rate by type of contact,

but that is not possible. Also not the other parameters. I have incorporated that

information into the text.

26. Line 240: I cannot extract that or at least not easily from Table 2, please be more precise?

I changed the sentence and included the calculations for the number: ”20 times”

higher in the result section under Analysis of quantity of isolation and

quarantine orders

27. Line 283: . . . or a reluctance of cases to give information about contacts

Thank you for the good addition.

28. Line 295: an update has been published, however, a lot of the evidence stems from SARS and

MERS; there is other publications with a focus on C-19, such as Pozo-Martin (EJE, 2023,

”Comparative effectiveness...”), Craig (JMIR 2021, ”Effectiveness of contact tracing...”) and

Fetzer (PNAS, 2021, ”Measuring the scientific effectiveness...”)

Thank you for the update. I included the Review bei Pozo-Martin.

29. Line 299: I dont understand what is meant here

I updated the sentence, and also the corresponding part in the limitation section.

30. Line 327: “The rough estimate for the median delay of 4 days that we found must be

considered as a flaw in the contact tracing. For COVID-19, an early detection of contact

persons is key to hinder transmission of the disease” – please check English (particularly

“flaw” and “hinder” ??)

Changed the two sentence - hopefully for the better

but did adherence increase and/or administrative feasibility improve?

Our study was not meant to measure adherence. We see that the number of non

contained cases goes up the shorter the quarantine period. For measuring

adherence we would needed to question cases and contacts after the order. I

would have loved to do that - also to see how the agency could have improved its

work. This would be something for another study.

32. Line 357: for modellers that is good information. however, again, most informative would be

the proportion of cases that were isolated by day in relation to symptom onset (see above).

I agree, but we dont have the symptom onset in our dataset.

33. The authors calculated the number of contacts per case. I think it would be more instructive

to stratify by those cases who were not previously in the system as contact person and those

who were. The latter will likely have very few contacts, the first might have a number of

contacts. In addition it would be nice to stratify by age (of the case; very broadly, say ¡18 and

18+ year old), because children should have quite a bit more contacts than adults.

You are perfectly right, this would be valuable information. But as stated in the

limitation section we cannot directly link a contact person to a case, because this

is information was not reliably stored in the software. We get a hint by the high

number of contact persons within the group of children, which is very likely due

to school.

Points raised by Reviewer No. 4

Thank you for your valuable feedback

1. It will help the readers understand the context better if the authors can also state in the

Abstract the total number of individuals analyzed.

Agreed. Added the figure of the total population

2. In Table 2, it will be helpful to mark the percentages as “%”, and provide SD for mean values.

The abbreviations are also a bit hard to understand at first glance (for example it is not

intuitive to me how “ncc” refers to “quarantines that had a directly following isolation

period”). As these abbreviations are used only once in Table 2, the authors can consider write

them out in words or improve the labels, if possible.

Thanks for the feedback. I Added the % and explained that ”ncc” means

non-contained case, which was indeed not intuitive. The table is to wide to fit the

page. I dont feel that the sd for the mean values are more important than the

other given values.

3. It is generally recommended to avoid using the term “elderly”. I suggest changing it to “older

adults”, “older persons”, or similar throughout the manuscript.

Thanks for the correction. I changed it to senior citizens

Points raised by Reviewer No. 5

1. I noticed that reference 21 was not cited in the manuscript. Additionally, please renumber the

references in the order they appear in the manuscript. As an example, references 36-38

currently appear in the first paragraph of the Introduction, which requires revision for proper

numbering.

Thank you for pointing that out. I changed the reference order

5/62. Line 11-12: ”Vaccines have been developed and proven successful in preventing severe disease

and death [40], ....” → I suggest change this sentence to ”Vaccines have been developed and

proven successful in inducing seroconversion (cite the following reference 1), preventing severe

disease and death [40], ....” Reference: [1] Safety and Seroconversion of Immunotherapies

against SARS-CoV-2 Infection: A Systematic Review and Meta-Analysis of Clinical Trials.

Pathogens. 2021 Nov 24;10(12):1537. doi: 10.3390/pathogens10121537. PMID: 34959492;

PMCID: PMC8706687.

Added the publication from Kevin Sheng-Kai Ma, as you suggested, although I

am not quite sure if it adds value.

3. Line 51-52: ”Jian et al. reported on the Taiwanese digital system TRACE analysing 487 cases

and 8051 contact persons [12].” → To emphasize the public health implications of these

particular systems, I suggest change this sentence to: ”Jian et al. reported on the Taiwanese

digital system TRACE, analyzing 487 cases and 8051 contact persons [12]. Public health

implications of these systems included surveillance of travel history (cite the following

reference 1) and ensuring an adequate quantity of personal protective equipment (cite the

following reference 2).” References: [1] Integrating travel history via big data analytics under

universal healthcare framework for disease control and prevention in the COVID-19 pandemic.

J Clin Epidemiol. 2021 Feb;130:147-148. doi: 10.1016/j.jclinepi.2020.08.016 [2] Big

Data-driven personal protective equipment stockpiling framework under Universal Healthcare

for Disease Control and Prevention in the COVID-19 Era. Int J Surg. 2020 Jul;79:290-291.

doi: 10.1016/j.ijsu.2020.05.091

Added the two publications from Kevin Sheng-Kai Ma. But as above I dont see

an added value.

---

## [Decision Letter · Decision Letter 3]

9 Feb 2024

COVID-19 isolation and quarantine orders in Berlin-Reinickendorf (Germany): How many, how long and to whom?

PONE-D-22-19240R3

Dear Dr. Schumacher,

We’re pleased to inform you that your manuscript has been judged scientifically suitable for publication and will be formally accepted for publication once it meets all outstanding technical requirements.

Kind regards,

Emanuele Crisostomi, PhD

Academic Editor

PLOS ONE

Additional Editor Comments (optional):

Reviewers' comments:

Reviewer's Responses to Questions

**Comments to the Author**

1. If the authors have adequately addressed your comments raised in a previous round of review and you feel that this manuscript is now acceptable for publication, you may indicate that here to bypass the “Comments to the Author” section, enter your conflict of interest statement in the “Confidential to Editor” section, and submit your "Accept" recommendation.

Reviewer #3: All comments have been addressed

Reviewer #4: All comments have been addressed

2. Is the manuscript technically sound, and do the data support the conclusions?

Reviewer #3: Yes

Reviewer #4: Yes

3. Has the statistical analysis been performed appropriately and rigorously? 

Reviewer #3: N/A

Reviewer #4: Yes

4. Have the authors made all data underlying the findings in their manuscript fully available?

Reviewer #3: No

Reviewer #4: Yes

5. Is the manuscript presented in an intelligible fashion and written in standard English?

Reviewer #3: Yes

Reviewer #4: Yes

6. Review Comments to the Author

Reviewer #3: In the point-by-point answer to the reviewers the authors have not inserted in their answer the updated text including – using the track changes mode – the indication of what has changed. If you just say for example “we incorporated the suggestion of the reviewer” the reviewer has to look for it himself or herself. Even though there is a version of the total manuscript with tracked changes this means double work for the reviewer because he/she has to search again for the position in the text and identify what has changed. Please, when you write a paper next time and get a review, always include in your point-to-point answer not only your answer but also the respective passage in the text of the manuscript.

Overall the paper has improved since the modification.

I still have a few comments:

I think you need to rethink the structure of the paper somewhat. It is unclear what the objective was, “quantification and analysis” cannot be a goal, it is a method, that is used for a purpose. You say in your response that you think it is important to know “if local agencies follow or dont follow national recommendations”. Why don’t you use that as a goal? something like “During the pandemic the burden of work for local health departments was enormous and the federal public health agency (the Robert Koch Institute) adapted its recommendation several times. We aim to investigate if a local health department was capable to follow the recommendations of the Robert Koch Institute.” With that goal in mind you need to add something in that regard in the abstract and come to a conclusion. To do that you can use a lot of the results that are already in the main part of the paper. A second goal is the estimation of the impact of the efforts of the health department.

I say something to both goals:

If you want to compare the number of RKI-recommended days of isolation or days of quarantine with the number of isolation / quarantine days per person I suggest to put both together in one graph, e.g. as a modification of figure 2 (for example the number of recommended days in a different colour or as a horizontal line), otherwise the reader has to pull it together from several locations (Table 1 and Figure 2) in the paper which is bothersome for the reader.

Regarding the impact I suggest to add to measures/indicators:

(1) the ratio of ncc : nncc which is roughly 4:1 in children and 10:1 in adults, or, the proportion of cases arising from known contacts that were contained (ncc/(ncc+nncc), which is about 80% in children and about 90% for adults.

(2) Another indicator for impact would be ncc/ni, which is the proportion of all cases (with isolation order) who were contained. This is roughly 28% in young children and goes down to 11% in adults.

(3) And a third indicator would be the estimation of the proportion of all infectious days in the population that you had prevented. That indicator combines data of quarantine and isolation. How can you do it:

Step 1: you estimate the number of infectious days (inf-days) in the population which is: ni*inf-days. You have to make an assumption about the infectious period, say day of illness onset -2 until day of illness onset + 5 or so (see e.g. Ke, Nature Microbiology, 2022, or: National Institute for Infectious Disease in Japan (“Active epidemiological investigation on SARS-CoV-2 infection…”). (Nota bene: there is very scarce literature on the shedding kinetic of virus that can be isolated, whereas shedding of virus that can be detected via PCR is abundant, but much different. What you need is the shedding of virus that can be isolated in cell culture)). But you could say you assume illness onset + 5 days (or something else). At any rate this is your denominator.

Step 2: You estimate the number of inf-days prevented. Because ni=ncc + rest, you have to calculate the number of prevented days first for the ncc and then for the “rest” (=ni-ncc). For the ncc the number of prevented inf-days is simply = ncc*inf-days.

For the rest (ni-ncc) the number of prevented inf-days you can calculate as follows:

you calculate the average day after symptom onset when you placed the isolation order. For example it is day 2 after symptom onset. Then the average number of prevented inf-days per person would be 3 days. Then the number of prevented inf-days is: (ni-ncc)*3 days.

Now you have to add the two numbers of prevented inf-days: ncc*inf-days + (ni-ncc)*3 days. This is the numerator.

Step 3: put both in relation: number of prevented inf-days = numerator / denominator.

(This calculation assumes that non-isolated cases would continue to meet their contacts as before their infection; that is a limitation)

I think you said that the day of symptom onset is not part of the database, but you should be able to link it with the cases-database where you have the names and the date of symptom onset. If you cannot link the isolation/quarantine database with the cases database for some reason you can still either take a best guess from your co-workers (“Delphi method”), and/or pull the files of say 100 cases, and look it up. Then you calculate what you need based on these 100 cases.

- I suggest you rephrase your title, for example: “Adherence and impact estimation of COVID-19 isolation and quarantine orders in Berlin-Rheinickendorf, Germany, 2020-2022”

- another small point: Although 2 contacts per case is probably too low Reference 22 (Mossong) is in my eyes not a good comparator for the number of contacts because for sure the population changed their behavior for an extended period of time during 2020-2021.

- Table 2 is doubled.

Reviewer #4: I would like to thank the authors for their thoughtful response to my previous comments. The paper is much improved, and I have no further questions.

7. PLOS authors have the option to publish the peer review history of their article (what does this mean?). If published, this will include your full peer review and any attached files.

Reviewer #3: No

Reviewer #4: No

---

## [Editor Report · Acceptance letter]

27 Feb 2024

PONE-D-22-19240R3 

PLOS ONE

Dear Dr. Schumacher, 

I'm pleased to inform you that your manuscript has been deemed suitable for publication in PLOS ONE. Congratulations! Your manuscript is now being handed over to our production team.

Kind regards, 

on behalf of

Professor Emanuele Crisostomi 

Academic Editor

PLOS ONE